# Air pollution exposure, respiratory consequences, and perceptions among urban African children living in poor conditions – A case study in Abidjan, Côte d'Ivoire

**Auriane Pajot** [1]*, **Marie Yapo**[2], **Sarah Coulibaly**[2], **Madina Doumbia**[3], **Sylvain Gnamien**[2], **Kouassi Kouao**[2], **Stéphane Ahoua**[2], **Sonia Adjoua Dje**[4], **Cathy Liousse** [5], **Raoul Moh**[6], **Joanna Orne-Gliemann** [1], **Flore Dick Amon Tanoh**[4], **Olivier Marcy** [1◉], **Véronique Yoboue**[2◉]

1 University of Bordeaux, Inserm U1219, IRD EMR 271, Centre de recherche Bordeaux Population Health, Bordeaux, France, 2 University Félix Houphouët-Boigny de Cocody, UFR des Sciences des Structures de la Matière et de Technologie, Laboratoire des Sciences de la Matière, de l'Environnement et de l'énergie Solaire, Abidjan, Côte d'Ivoire, 3 Universty Peleforo Gon Coulibaly, Korhogo, Côte d'Ivoire, 4 Service de Pédiatrie, CHU d'Angré, Abidjan, Côte d'Ivoire, 5 Université Toulouse III Paul Sabatier, Observatoire Midi-Pyrénées, Toulouse, France, 6 Programme PACCI-ANRS Research Site, Abidjan, Côte d'Ivoire

◉ These authors contributed equally to this work.
* auriane.pajot@u-bordeaux.fr

## Abstract

Air pollution can severely impact child lung health but is often not considered a public health priority by policy-makers and population in low-and-middle income countries. We conducted an interdisciplinary mixed method study to assess exposure to air pollution and respiratory health on children aged 5-10 living in poorly condition in Yopougon, a district of Abidjan, Côte d'Ivoire, and to evaluate parent and child perceptions and knowledge of air pollution. We measured pollution exposure with indoor and outdoor $PM_{10}$ and $PM_{2.5}$ concentrations and questionnaires, assessed children's respiratory health with ISAAC questionnaire, clinical evaluation, spirometry or RINT, depending on their ability to perform a forced expiration, their perception of air pollution with a "Draw and express yourself" activity and that of parents with semi-structured interviews and questionnaires. We enrolled 124 children from 65 households, that used mixed cooking with gas and charcoal in settings with important environmental air pollution. Median 48-hour $PM_{10}$ and $PM_{2.5}$ concentrations were 126.7 (IQR: 82.7) and 60.8 (IQR: 50.7) μg/m³, indoor respectively, and 113.4 (IQR: 64.2) and 58.2 (IQR: 36.9) μg/m³, outdoor in courtyards. 21 (16.9%) children reported wheezing in the previous year, 65 (52.4%) reported dry cough at night, and 63 (72.4%) had lung function impairment on spirometry with 24 (27.6%) asthma, 19 (21.8%) non reversible obstruction and 20 (23.0%) restrictive pattern. Adults and children were able to identify visible sources of air pollution but largely ignored effects on health. Despite high exposure to air pollution with particulate matter concentrations significantly exceeding WHO recommendations, and a high prevalence of respiratory symptoms, lung function

**Data availability statement:** Study data will not be publicly available. Data could be made available by study group to any researcher interested. Deidentified participant data and a data dictionary can be made available and shared under a data transfer agreement. Requests for access to the Sepol-CI study data should be sent to auriane.pajot@u-bordeaux.fr.

**Funding:** Sole funder : This study received financial support from the French government in the framework of the University of Bordeaux's France 2030 program / GPR IPORA - Interdisciplinary Policy-Oriented Research on Africa https://ipora.africa/fr/ The funders had no role in study design, data collection and analysis, decision to publish, or preparation of the manuscript.

**Competing interests:** The authors have declared that no competing interests exist.

impairment and asthma, among children, children and adults, perception of air pollution as a health issue was very limited. Recommendations and awareness-raising for parents and children, starting at primary school, are needed to limit the exposure to air pollution and its respiratory consequences.

## Introduction

Air pollution is a growing environmental concern, and a major global health threat. According to the World Health Organization (WHO), exposure to ambient (outdoor) and household (indoor) air pollution causes 6.7 millions premature deaths each year, and the loss of millions of years of healthy life [1]. In 2019, 99% of the world's population lived in areas where air pollutant concentration exceeds WHO recommended air quality thresholds [1]. In adults, air pollution is responsible for chronic respiratory diseases such as chronic obstructive pulmonary disease (COPD) and lung cancer, as well as cardiovascular and cerebrovascular diseases [2–4]. In 2019, 89% of premature deaths due to outdoor pollution occurred in low- and middle-income countries (LMIC) [1].

Populations from LMIC are largely exposed to air pollution, particularly household pollution caused by the use of low-cost, accessible biomass fuels for cooking and heating [5–9]. By 2019, almost 3 billion people were using biomass fuels such as wood, coal, animal dung, agricultural residues and oil to cook, heat and light their homes [10]. In 2019, only 19% of the population in the African region had access to clean energy, compared to 90% in the European and American regions [11]. Using biomass fuels in poorly ventilated homes, combined with inefficient cooking methods, exposes people to Particulate Matter (PM) concentrations that are sometimes 100 times higher than the thresholds recommended by the WHO [10,12]. PM include microscopic matter suspended in the air, mainly resulting from the use of biomass as a fuel. These are fine particles with diameters of less than 10 μm ($PM_{10}$) and 2.5 μm ($PM_{2.5}$) [13]. Inhaled in large quantities, they increase cardiopulmonary morbidity and mortality [14].

Children are particularly vulnerable to pollution as they are more sensitive to atmospheric pollutants [15–17]. Their organs are immature, and they have a higher respiratory rate than adults, inhaling more air and therefore more $PM_{2.5}$, $PM_{10}$ and other pollutants responsible for consequences on lung development, limitation of pulmonary function, respiratory infections and aggravation of asthma [2,3,15–17]. In 2021, more than 700,000 deaths among children under 5 years of age were attributed to diseases related to air pollution [18]. Exposure to household air pollution, particularly from burning biomass in open fireplaces, increases the risk of pneumonia [10,19–21], of cough accompanied by reported rapid breathing or difficulty breathing on children [22–28] and low birth weight [29,30]. Children under five in low-income countries face a risk of death from pollution exposure that is at least 60 times higher than in high-income countries [31].

Despite the high burden of both non-communicable and communicable diseases due to air pollution, these are rarely considered as public health priorities in low and middle-income countries, still facing the heavy burden of pandemic infectious diseases such as HIV, malaria, and tuberculosis. Pollution legislation is weak or unenforced. For many sub-Saharan African countries, there is insufficient ambient air quality monitoring and a lack of capacity to collect population health data, which hinders evidence-based interventions [32,33]. People in low- and middle-income countries are more exposed to pollution, especially the poorest populations [34]. These populations also tend to have limited knowledge about air pollution [33,35].

In Côte d'Ivoire, outdoor air pollution is mainly due to anthropogenic activities such as burning of dumped waste in open air, biomass use for domestic cooking and fish smoking

activities [7,36,37]. With increasing urbanization, unregulated industries activities and urban sprawl, green spaces are declining, and homes are being built close to industries, exposing people to pollution. The use of outdated vehicles, with an old fleet of vehicles using mostly low-quality fuel, is one of the main sources of outdoor air pollution. Urban air in Côte d'Ivoire is also full of dust in suspension due to the poor quality of the roads [6,38,39]. Several studies carried out between 2005 and 2020 showed that $PM_{2.5}$ concentration levels were 3 to 15 times higher than WHO recommended thresholds, with the highest levels found in low-income neighborhoods [6,40]. As part of the CHAIRPOL (Urban air pollution and non-communicable diseases in ecohealth) program, a study carried out in 2016-2017 in poorly living condition households of Yopougon, a densely population area of Abidjan, revealed high biomass fuel use, elevated PM2.5 levels, and asthma-like symptoms in children [41].

We sought to describe exposure to air pollution, respiratory health, and to understand the perception of pollution in a disadvantaged urban environment in Abidjan, Côte d'Ivoire, among children aged 5 to 10 and their parents. This interdisciplinary work could contribute to a better understanding of air pollution issues and find concrete solutions to protect exposed populations and improve their quality of life.

## Methods

### Study design

The SEPol-CI study was a multidisciplinary cross-sectional study with enrollments conducted between June 23 and August 25, 2022, during the rainy season. It involved various research teams in public health/epidemiology, atmospheric physics and chemistry, sociology, pediatric pulmonology, focusing on children aged 5 to 10, living in disadvantaged urban environments. The study used quantitative and qualitative methods, including measurements of $PM_{10}$ and $PM_{2.5}$ inside and outside the children's homes, exposure to air pollution and knowledge questionnaires, the standardized International study of asthma and allergies in childhood (ISAAC) questionnaires on asthma and a questionnaire on respiratory health, clinical assessment and respiratory function tests among children, interviews with children and their parents about air pollution perceptions and the "Draw and express yourself" activity for children.

### Study setting

We carried out the study in Andokoi, a district of Yopougon, the most densely-populated suburb of Abidjan, capital city of Côte d'Ivoire. Andokoi is a working-class neighborhood located North of Yopougon, on the edge of the Banco forest, close to an industrial zone [41]. Proximity to the protected forest encourages local residents to exploit these resources illegally with intensive logging, and harvesting of various forest products [41]. The majority of men work in the industrial zone, while the women have informal jobs and activities as shopkeepers in the surrounding area [41]. Andokoi is one of Yopougon's main garbage dumping and septic tank emptying sites [42]. We selected Andokoi for this study, as exposure to pollution and its impact on respiratory health had been studied previously in this neighborhood during the CHAIRPOL program in 2016-2017 [41].

### Study population and sample size

We enrolled children aged between 5 and 10 years with informed consent signed by a parent or guardian and an assent signed for children over 7 years. Assuming 2 children per household in the 5-10 age group, the sample size was based on a convenience sample of around 100 children. A household was defined as all persons living under the same roof. A sub-sample

of 10 adults with at least one child included in the study was targeted for the qualitative interviews.

## Study procedures

Data was collected from June to September 2022. We based ourselves on the list of households that had participated in the previous study (Kouao, 2020) [41]. A number of new households were also included base on willingness to participate to complete our population in the same area of the previous study. Data from questionnaires were collected on a digital tablet by an investigator.

Data on sources of exposure, characteristics of houses and living environments, were collected from questionnaires, developed by a multi-disciplinary team on the basis of the questionnaire constructed 5 years ago during the study (Kouao, 2020) [41]. This data was supplemented by measurements of PM concentrations. Concentrations of $PM_{2.5}$ and $PM_{10}$ were measured every two minutes over a 48-hour period, using two Aéroqual sensors located inside and outside homes. Measurements of PM concentrations were retrieved from the database and analysed by aerology students from the LASMES laboratory in Abidjan. To ensure proper calibration of the devices and consideration of environmental conditions during pollution measurements, correction factors have been applied to the concentrations.

Data from the children's health questionnaire included the first part of the standardised ISSAC questionnaire (International study of asthma and allergies in childhood), the list of respiratory symptoms and signs and the history of respiratory illness. Health questionnaire was supplemented by respiratory function tests, and a physician's recording of clinical signs and vital parameters. Respiratory function tests were carried out by a medical doctor, spirometry measurements were taken in children, and airway interruption resistance (Rint) were performed on children who were unable to perform spirometric curves. It is a non-invasive test, easily performed on young children who are not able to produce the forced expirations required for spirometry. Vital parameters were measured by a nurse.

Semi-structured interviews were conducted with the parents of the children included in the study to supplement the data collected by questionnaire and to understand individuals' perceptions of air pollution. The interviews were conducted according to an interview guide and recorded by a trained interviewer, in a space away from homes to facilitate discussion and privacy.

The perception of air pollution among adults was surveyed using questionnaires and semi-structured interviews with a smaller sample of parents. The "Draw and express yourself" activity was used to understand children's perceptions of air pollution [43]. In this method, children are asked to draw on a specific topic on a blank sheet of paper and then debriefed on what they have drawn, supplementing these explanations with semi-directive questions. The "draw-then-explain" method, has already been carried out in Morocco with children aged 7 to 12 [43].

## Main respiratory health outcomes and definitions

The primary respiratory outcome was defined as the presence of respiratory symptoms suggestive of asthma and/or lung function impairment (LFI). We considered as respiratory symptoms suggestive of asthma defined by the ISAAC questionnaire: presence of wheezing, history of asthma attack, sleep disturbed by wheezing, speech limited, wheezing in the chest during or after exercise, dry cough at night, in the last 12 months [44]. Pulmonary function assessments were performed using spirometry and Rint, measured with a Spirodyn pneumotachograph.

LFI was defined by the presence of spirometric abnormalities characterized by either obstructive (OVD), restrictive (RVD) or mixed ventilatory disorders. Definitions of LFI on children are not as clear-cut as on adults [45,46]. A non-reversible OVD was defined by a forced expiratory flow at 25-75% of forced vital capacity (FEF25-75) < 70% of the theoretical value and/or forced expiratory volume in one second (FEV1) < 80% of the theoretical value and non-reversible on the bronchodilator reversibility test. A RVD was defined by a forced vital capacity (FVC) < 80% and normal FEF25-75 or FEV1, respectively ≥70% et ≥80%. A mixed disorder was characterized by FVC < 80%, FEF25-75 < 70% and or FEV1 < 80%. Asthma was defined by a reversible OVD with a FEV1 greater than 12% of baseline on post-test reversibility to bronchodilatator. In children unable to undergo a full spirometry, OVD on Rint was defined as a Rint > 150% of the theoretical value with reversibility of obstruction objectified by a decrease in Rint ≥ 35% of the predicted value after reversibility testing.

## Other health outcomes

Children's health data included: clinical signs on auscultation by a physician, respiratory symptoms (daily cough, throat irritation) and history, vital parameters, spirometric parameters, and Rint measurements. The spirometric parameters collected were: FVC, FEV1, Tiffeneau's ratio (FEV1/FVC) and FEF25-75. The Rint parameter collected was specific airway resistance in kPa/l/s (sRaw), which measures total pulmonary resistance [47].

## Pollution exposure and perceptions outcomes

Exposure to air pollution was characterized by sources of pollution exposure, housing characteristics, and average daily household and outdoor $PM_{2.5}$ and $PM_{10}$ concentrations. Perceptions of air pollution among children and adults included: understandings of air pollution sources, health and environmental consequences, solutions, information and prioritisations to the problem.

## Data analysis

**Quantitative data analysis.** We describe dwelling and household characteristics and potential sources of pollutants in the home and related to lifestyles, children's demographic and medical data, respiratory function test data and adults' perceptions and knowledge of air pollution. We averaged $PM_{2.5}$ and $PM_{10}$ data collected households and outdoors over 48 hours to obtain a median value.

We conducted binary logistic regression to identify an association between potential risk factors for the following binary dependent variables: LFI at spirometry, asthma at spirometry, and wheezing in the last 12 months, and all 3 grouped together VS healthy children. We used Chi-square tests, Fisher's exact tests, Wilcoxon tests and Kruskal-Wallis tests to select independent variables with a P value <0.20, which were then included in a logistic regression model.

The explanatory variables tested included sex, age, body weight, height, floor material (cement/tiles), kitchen type, home tabacco smoker, fish smoking activity in the household, presence of pets at home, use of spray and aerosol, type of fuel, type of cooking equipment for cooking, household and outdoor $PM_{2.5}$ and $PM_{10}$. Statistical analysis was performed using the R-Studio software.

**Analysis of qualitative data.** Thematic summaries were produced after transcribing the interviews and carrying out a thematic analysis following the themes predefined in the interview guide and allowing the salient themes to emerge from the data. Selected interview extracts and quotes were translated into English for the purpose of this publication; the accuracy of the translation was verified.

### Ethical approval

The study protocol has been approved by the Côte d'Ivoire National Ethics Committee for Health and Life Sciences (CNESVS). In addition, the authorizations to launch the project were obtained from the local chiefdom.

## Results

### Population and living environment characteristics

We included 124 children from 65 households, of median age 7 years (IQR: 6 – 9), with 61 (49.2%) girls (Fig 1, Table 1). 47 children from 29 households had participated in the previous study (Kouao, 2020) [41]. An average of 7 persons (IQR: 5 – 10) lived in each household, 30 of them in a 1 or 2-room homes (46.2%) and 20 in 3-room homes (30.8%). 62 households lived in shared courtyards (95.4%). Flooring was cement for 44 households (67.7%) and tiles for 21 (32.3%). For 41 homes, the kitchen was located outside, in the yard (63.1%) and 10 homes had an indoor kitchen with no windows or ventilation system (15.4%). Sprays or aerosols such as mosquito repellents were used in 55 households (84.6%), and 16 households had a pet in the home (24.6%). 29 households reported practicing fish smoking activity in the backyard (44.6%), with 69% smoking few times a month and others, few times a year (Table 1).

### Exposure to air pollution

All households used gas as their daily cooking fuel. In addition, charcoal was used by 44 (67.7%) households for cooking. No household used charcoal as its sole fuel, 15 (23.1%) households used gas exclusively and 6 (9.2%) households used a mix of gas, charcoal and wood. In interviews parents reported using both gas and charcoal, used as a complement to gas, either for the time it takes to change the gas cylinder bottle, or for long-cooking dishes that consume a lot of gas. A traditional stove was used in 51 homes (78.5%) for charcoal and 3 households reported using also an improved stove (4.6%). In this study, the improved stove is distinguished from the traditional one by its combustion chamber, which limits smoke propagation, reduces cooking time, and decreases the amount of fuel used. The gas cooker is a butane gas cylinder with a pot support. For lighting cooking fires, 47 (94.0%) households said they used dry rubber; the second most common method was plastic (32.0%) (Table 1).

The median household $PM_{10}$ concentration over 48 hours, measured indoor 34 households, was 126.7 µg/m³ (IQR: 82.7) and the median outdoor $PM_{10}$ concentration, measured in the common courtyard of 28 households was 113.4 µg/m³ (IQR: 64.2). Corresponding $PM_{2.5}$ concentrations were 60.8 µg/m³ (IQR: 50.7) and 58.2 µg/m³ (IQR: 36.9), inside and outside the house, respectively (Table 1).

### Respiratory health

According to parents, 39 (31.4%) children had presented wheezing in the chest or noisy breathing during their lifetime and 21 (16.9%) during the previous year; 27 (21.8%) had already have an asthma attack, 65 (52.4%) had dry cough at night in the past 12 months, 36 (29.0%) complained of burning or watery eyes and 49 (39.5%) had a cough on a daily basis (Table 2). On the 124 children enrolled, 99 were examined by a physician, 20 (20.0%) had cough on the day of examination, 14 (14%) reported chest pain (14.0%), 1 (1.0%) exertional dyspnea and 10 (10.0%) had rhinorrhea (Table 2).

Overall, 87 (70.1%) children underwent a full spirometry with a test of reversibility to bronchodilators (salbutamol) and 12 (9.7%) a Rint measurement. LFI was identified in 63 children who underwent spirometry; 19 (21.8%) had non-reversible OVD, 24 (27.6%) reversible OVD

**Table 1. Characteristic of household dwellings and pollutant concentration (N = 65).**

| Characteristics | N if ≠ N* | N*= 65<br>N (%) or Median (Q1-Q3) |
|---|---|---|
| Number of occupants in house | | 7 (5–10) |
| Number of rooms | | |
| 1 or 2 | | 30 (46.2) |
| 3 | | 20 (30.8) |
| 4 or more | | 15 (23.0) |
| Common courtyard | | 62 (95.4) |
| Number of years lived in residence | | |
| 1 to 5 years | | 7 (10.8) |
| > 5 years | | 58 (89.2) |
| Floor material | | |
| Cement | | 44 (67.7) |
| Tiles | | 21 (32.3) |
| Soil | | 0 |
| Kitchen type | | |
| Closed, with window/screen | | 14 (21.5) |
| Closed, windowless | | 10 (15.4) |
| Unenclosed, in courtyard or corridor | | 41 (63.1) |
| Home tobacco smoker | | 6 (9.2) |
| Fish smoking activity in the household | | 29 (44.6) |
| A few times a month | 29 | 20 (69.0) |
| A few times a year | 29 | 9 (31.0) |
| Pets at home | | 16 (24.6) |
| Dog or cat | 16 | 14 (87.5) |
| Other: goats, mangosteen | 16 | 2 (12.5) |
| Use of sprays and aerosols in the home | | 55 (84.6) |
| Child near the stove during cooking times | | |
| Never | | 46 (71) |
| Sometimes | | 11 (17) |
| Often | | 7 (11) |
| Very often | | 1 (1) |
| Type of fuel used for cooking | | |
| Gas only | | 15 (23.1) |
| Gas and coal | | 44 (67.7) |
| Gas, coal and wood | | 6 (9.2) |
| Cooking fire ignition | | |
| Dry rubber only | 50 | 34 (68.0) |
| Dry rubber and plants or paper | 50 | 4 (8.0) |
| Dry rubber and plastic | 50 | 9 (18.0) |
| Plastic only | 50 | 3 (6.0) |
| Type of cooking equipment for cooking | | |
| Gas cooker only | | 13 (20.0) |
| Gas cooker and traditional oven | | 49 (75.4) |
| Gas cooker, traditional oven and Improved cooker | | 2 (3.1) |
| Gas cooker and improved oven | | 1 (1.5) |
| Indoor particulate matter concentration | | |
| $PM_{10}$ μg/m³ | 34 | 126.70 (95.91–178.57) |

*(Continued)*

**Table 1.** (Continued)

| Characteristics | N if ≠ N* | N*= 65 N (%) or Median (Q1-Q3) |
|---|---|---|
| PM$_{2.5}$ μg/m³ | 34 | 60.85 (45.56–96.24) |
| Outdoor particulate matter concentration | | |
| PM$_{10}$ μg/m³ | 28 | 113.45 (93.11–157.30) |
| PM$_{2.5}$ μg/m³ | 28 | 58.21 (45.14–82.01) |

(asthma) and 20 (23.0%) RVD (Table 2). No child had a mixed disorder. No LFI was identified in children who underwent a Rint. Among the children with an LFI, 28 (32.2%) had an abnormal FEV1 below the norm and 14 (16.1%) for FEF25-75. There was no significant correlation between spirometric abnormalities and respiratory symptoms and clinical signs (Table 3). Logistic regressions assessing factors associated with the presence of LFI in children were not statistically significant (results not presented in the article).

## Adult perceptions and knowledge of pollution

**Understandings of air pollution and its sources.** Of 59 adults surveyed, 40 (67.8%) qualified their neighborhood as affected by air pollution (Table 4).

We conducted semi-structured interviews with 5 women and 3 men. During interviews, respondents identified air pollution as bad smells, or the presence of smoke: *"No, it doesn't spoil the air, there are no bad smells"* (interview 4) or again, about air pollution in the neighborhood: *"... there are no worries, there are no bad smells... It's nice here..."* (interview 7). Furthermore, air pollution was often confused with pollution in general, and water and soil pollution in particular. In the questionnaire, factories and industries were identified by 49 (83.0%) adults as having a strong or very strong impact on air pollution, dust was cited by 47 (79.7%), vehicles and transport by 45 (76.3%), garbage burning by 41 (70.7%), fish-smoking activities by 34 (57.6%) and open combustion fires by 28 (47.5%). These sources of air pollution were also mentioned in the interviews, as were cigarettes.

Most adults interviewed cited as sources of air pollution, waste in the gutters, stagnant water circulating in the village and plastic waste: *"The waste there... it's a problem we don't know where to remove it when there's too much, it goes into the gutter"* (interview 8).

Gas is gradually replacing some of the use of coal for financial or practical reasons: *"We know that wood is expensive, so we use coal, but now coal is expensive too, so we use a bit of gas."* (interview 5), *"You know, in the courtyard, coal is just sometimes used for sauces, because sauces take a long time and use a lot of gas, but otherwise it's just gas, you light it and it's hot, now we use gas, coal stove is tiring and coal costs a lot of money..."* (interview 7).

**Health consequences of air pollution.** Overall, 33 (55.9%) adults reported being informed about air pollution in general, 38 (64.4%) about the effects of air pollution on health, and 16 (27.1%) about protecting oneself from air pollution. Cough and runny nose were symptoms most often cited as the health consequences of air pollution: *"It's especially true for children, after they cough a lot, it gives them colds and even tires them out a lot. But yes, especially children, but when you're older, it's okay"* (interview 5), *"My children cough at night"* (interview 8). Few respondents (13, 22.0%) felt that air pollution had little or no impact on their health.

Poor lifestyle habits and poor home maintenance were identified as the cause of air pollution by respondents: *"the air in the house is no longer aerated, it's no longer refined, so it becomes polluted"* (interview 3), *"As I said, it's good, we clean the house, we clean the yard and it's clean, there's no problem at home."* (interview 8), most often due to ignorance "it's

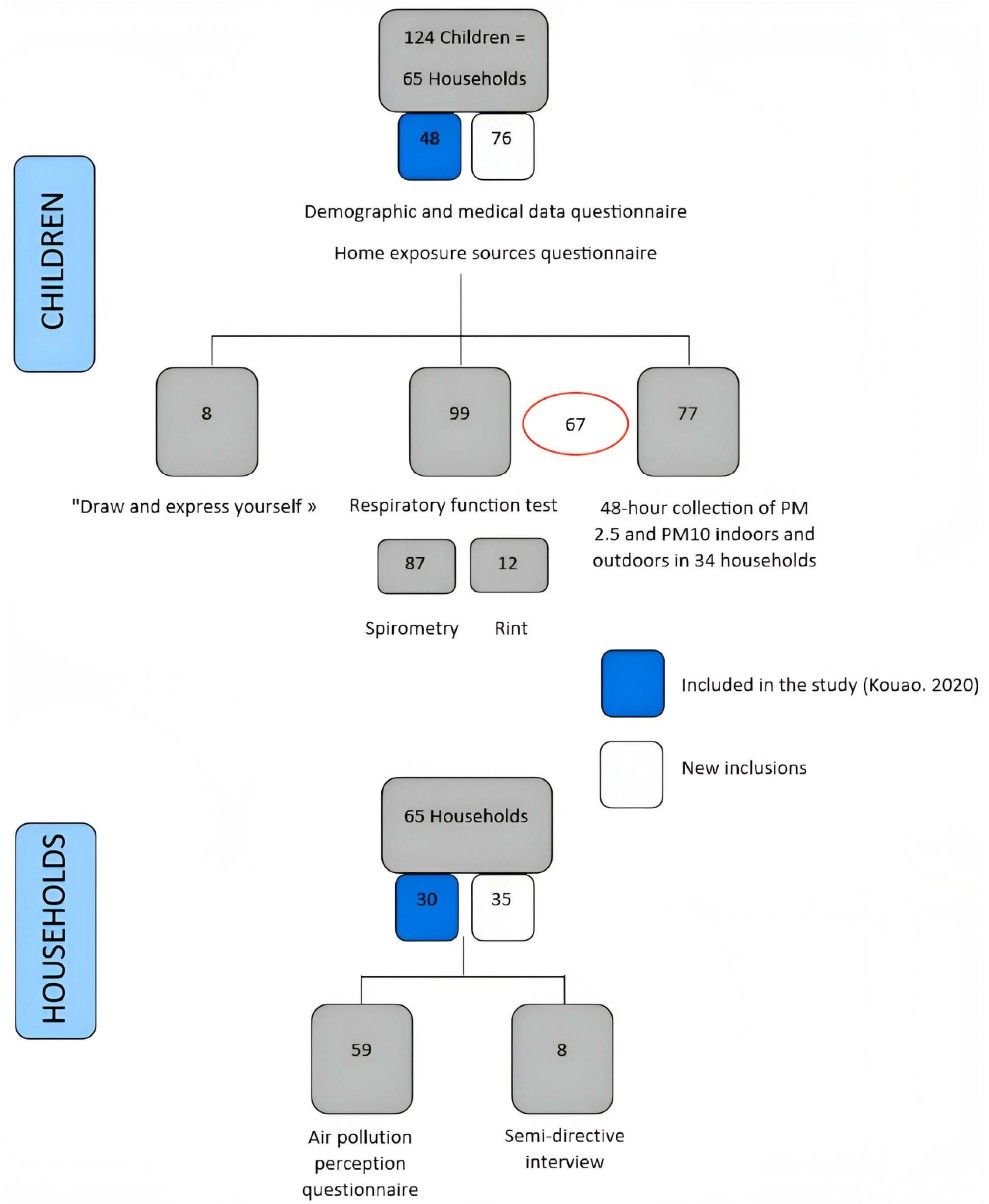

**Fig 1. Flow diagram of children and households included in the study, SEPol-CI, 2022, Andokoi.**

unconsciousness... there's literacy" (interview 1), *"I think it's because they don't know, they don't know the importance of air"* (interview 3).

**Environmental consequences of pollution.** In the questionnaire, 6 (10.2%) don't know air pollution effect on the environment. Environmental consequences of pollution were not mentioned in the interviews, but some cited the Banco forest as an important factor in air quality: "It's the Banco itself that means there's less pollution, there's the forest, the forest consumes gas, there's the ecosystem. It also brings us rain, not to mention the sea and the ocean. If this forest is destroyed, it's all over... it would be a catastrophe..." (interview 1), "and then we're right on the edge of the Banco forest, where the air is still much cleaner than inside the neighborhood over there" (interview 3).

**Table 2. Demographic and medical characteristics, clinical signs, vital parameters and prevalence of lung function impairment of children (N = 124).**

| Characteristics | N if ≠ N* | N* =124 N (%) or Median (IQR) |
|---|---|---|
| Gender Girls | | 61 (49.2) |
| Age (years) | | 7 (6–9) |
| Weight (kilo) | 99 | 24 (20–27) |
| Height (cm) | 99 | 123 (116–133) |
| **ISSAC Questionnaire** | | |
| Noisy breathing/whistling in the chest in the past | | 39 (31.4) |
| Noisy breathing/whistling in the chest (last 12 months) | 39 | 21 (53.8) |
| Wheezing attacks (last 12 months) | 21 | |
| None | | 1 (4.8) |
| 1 to 3 | | 14 (66.7) |
| 4 to 12 | | 5 (23.8) |
| More than 12 | | 1 (4.8) |
| Sleep disturbed by wheezing (last 12 months) | 21 | |
| Never | | 4 (19.0) |
| Less than one night a week | | 11 (52.4) |
| One or more nights a week | | 6 (28.6) |
| Speech limited to one or two words (last 12 months) | 21 | 14 (66.7) |
| Asthma attacks | | 27 (21.8) |
| Wheezing in the chest during or after exercise (last 12 months) | | 24 (19.3) |
| Dry cough at night (last 12 months) | | 65 (52.4) |
| **Questions asked outside of the ISAAC questionnaire** | | |
| Daily cough | | 49 (39.5) |
| Burning or watery eyes | | 36 (29.0) |
| Throat irritation | | 28 (22.6) |
| History of pneumonia | | 11 (8.9) |
| **Clinicals signs reported on the day of clinical examination** | 99 | |
| Cough | | 20 (20.0) |
| Chest pain | | 14 (14.0) |
| Effort dyspnea | | 1 (1.0) |
| Rhinorrhea | | 10 (10.0) |
| Heart rate (in 1 minute) | | 85 (74–92) |
| Breathing frequency (in 1 minute) | | 23 (20–26) |
| **Spirometry parameters** | 87 | |
| FVC in bronchodilatation pre-test | | 1.23 [0.98–1.53] |
| Abnormal FVC <80% theoretical value | | 60 (69.02) |
| FEV1 in bronchodilatation pre-test | | 1.24 [0.98–1.52] |
| Abnormal FEV1< 80% theoretical value | | 28 (32.2) |
| FEV1/FVC in bronchodilatation pre-test | | 100 [100–100] |
| Abnormal FEV1/FVC < 80% theoretical value | | 0 |
| PEF in bronchodilatation pre-test | | 3.89 [3.14–4.46] |
| Abnormal PEF <80% theoretical value | | 28 (32.2) |
| FEF25-75 in bronchodilatation pre-test | | 2.68 [2.38–3.40] |
| Abnormal FEF-25-75 < 70% theoretical value | | 14 (16.1) |
| Presence of LFI in spirometry | | 63 (72.4) |
| Non reversible obstructive pattern | | 19 (21.8) |

*(Continued)*

**Table 2.** (Continued)

| Characteristics | N if ≠ N* | N* =124 N (%) or Median (IQR) |
|---|---|---|
| Reversible obstructive pattern (asthma) | | 24 (27.6) |
| Restrictive pattern | | 20 (23.0) |
| Mixed pattern | | 0 |
| **Rint parameters** | 12 | |
| Rint in bronchodilatation pre-test | | 0.72 [0.56–1.01] |
| Abnormal Rint pre | | 1 (8.3) |
| Presence of LFI in Rint | | 0 |
| Reversible obstructive pattern (asthma) | | 0 |
| **Total LFI** | 99 | 63 (63.6) |

LFI = lung function impairment; FVC = forced vital capacity; FEV1 = forced expiratory volume in one second; PEF = peak expiratory flow; FEF25-75 = forced expiratory flow at 25-75%, Rint = Lung resistance due to iterative flow interruptions; IQR = interquartile range

**Solutions to the problem of air pollution exposure.** Solutions to the problem of air pollution are individual-level behavior changes. The notion of keeping one's environment clean was very often mentioned: *"It's cleanliness that counts, and that's enough. Houses have to be clean."* (interview 1), *"Keeping the house clean is normal, you have to keep your house clean, I sweep the front yard every day..."* (interview 7). Respondents suggested raising public awareness to start addressing the air pollution problem: *"We're going to raise awareness in the neighborhood so we can do a better job of all this"* (interview 6). They felt that it's also up to political leaders to take action to fight the problem: *"It's up to the politicians, not us, they have the money, they know what needs to be done, to find solutions..."* (interview 5).

**Information on and prioritisation of pollution prevention.** Solutions to the problem based on better knowledge that would be coming from more education and information. Participants explained that most of the information on pollution and health came from watching television, but that overall poor access to education and information on these topics led to limited knowledge: *"it's unconsciousness... there's literacy"* (interview 1), *"I think it's because they don't know, they don't know the importance of air"* (interview 3). Poor lifestyle habits and poor home maintenance were identified as the cause of air pollution: *"the air in the house is no longer aerated, it's no longer refined, so it becomes polluted"* (interview 3), *"As I said, it's good, we clean the house, we clean the yard and it's clean, there's no problem at home."* (interview 8).

**Public interest.** Air pollution is not a source of concern or interest for the population: *"They don't care at all, because if they did, everyone would make an effort to ensure good air quality. But they don't give a damn, it's how they're going to live from day to day that interests them, otherwise the rest..."*. (interview 3), *"We often live here, so we look at things but don't react, you have to manage, so you don't think about it, you don't have to think about it because you have no choice, there are other concerns for the family"* (interview 5), *"Well, as I told you, I've lived through it, so it doesn't worry me, it's not something that prevents me from living or breathing. There are other things to think about"* (interview 6). The interviewees explained that the population faces many other problems that take priority over air pollution: *"So they're not worried about how they're going to earn a living so they can eat, that's all, if not how to improve the air, they don't give a damn"* (interview 3), *"You know, it's hard here, we're here but we don't know for when because it's expensive and you have to work and even if you work, your child is sick and you can't always find the medicine or the hospital"* (interview 7). Data on the perception of air pollution in adults are presented in Table 4.

**Table 3. Comparative data on demographic and medical characteristics, clinical signs and vital parameters between children with a spirometry lung function impairment on spirometry and children without these characteristics (N = 99).**

| Characteristics | N* = 99 Overall | N* = 36 Healthy Children | N*= 63 Children with LFI | P-value² |
|---|---|---|---|---|
| Gender Girls | 51 (51.5) | 18 (50.0) | 33 (52.4) | 0.8 |
| Age (years) | 7 (6-9) | 7 (5-8) | 8 (6-9) | 0.023 |
| Weight (kilo) | 24 (20-27) | 23 (18-26) | 25 (20-27) | 0.11 |
| Size (cm) | 123 (115-133) | 121 (111-128) | 126 (120-134) | 0.024 |
| **Part I ISAAC questionnaire** | | | | |
| Noisy breathing/whistling in the chest in the past (ISAAC) | 36 (36.4) | 14 (38.9) | 22 (34.9) | 0.7 |
| | N=36 | N=14 | N=22 | |
| Noisy breathing/whistling in the chest (last 12 months) | 19 (52.8) | 9 (64.3) | 10 (45.4) | 0.3 |
| Wheezing attacks (last 12 months) | N=19 | N=9 | N=10 | 0.7 |
| None | 1 (5.3) | 0 | 1 (10.0) | |
| 1 to 3 | 12 (63.2) | 7 (77.8) | 5 (50.0) | |
| 4 to 12 | 5 (26.3) | 2 (22.2) | 3 (30.0) | |
| More than 12 | 1 (5.3) | 0 | 1 (10.0) | |
| Sleep disturbed by wheezing (last 12 months) | N=19 | N=9 | N=10 | 0.8 |
| Never | 3 (15.8) | 2 (22.2) | 1 (10.0) | |
| Less than one night a week | 10 (52.6) | 4 (44.4) | 6 (60.0) | |
| One or more nights a week | 6 (31.6) | 3 (33.3) | 3 (30.0) | |
| | N=19 | N=9 | N=10 | |
| Speech limited to one or two words (last 12 months) | 13 (68.4) | 6 (66.7) | 7 (70.0) | >0.9 |
| Asthma attacks | 25 (25.3) | 10 (27.8) | 15 (23.8) | 0.7 |
| Wheezing in the chest during or after exercise (last 12 months) | 21 (21.2) | 7 (19.4) | 14 (22.2) | 0.7 |
| Dry cough at night (last 12 months) | 47 (47.5) | 16 (44.4) | 31 (49.2) | 0.6 |
| **Questions asked outside of the ISAAC questionnaire** | | | | |
| Daily cough | 36 (36.4) | 14 (38.9) | 22 (34.9) | 0.7 |
| Burning or watery eyes | 29 (29.3) | 9 (25.0) | 20 (31.7) | 0.5 |
| Throat irritation | 25 (25.2) | 8 (22.2) | 17 (26.9) | 0.6 |
| History of pneumonia | 10 (10.1) | 6 (16.7) | 4 (6.3) | 0.2 |
| History of tuberculosis | 0 | 0 | 0 | |
| **Clinicals signs reported on the day of clinical examination** | | | | |
| Cough | 20 (20.20) | 6 (16.7) | 14 (22.22) | 0.5 |
| Chest pain | 14 (14.14) | 7 (19.4) | 7 (11.11) | 0.3 |
| Effort dyspnea | 1 (1.0) | 1 (2.8) | 0 | 0.4 |
| Rhinorrhea | 10 (10.1) | 1 (2.8) | 9 (14.3) | 0.088 |
| Heart rate (in 1 minute) | 85 (74–92) | 85 (72–96) | 85 (74–92) | >0.9 |
| Breathing frequency (in 1 minute) | 23 (20–26) | 23 (20–26) | 24 (20–26) | 0.9 |
| **Spirometry parameters** | **N=87** | **N=24** | **N=63** | |
| FVC in bronchodilatation pre-test | 1.23 (0.98–1.53) | 1.45 (1.27–1.75) | 1.11 (0.93–1.43) | <0.001 |
| FEV1 in bronchodilatation pre-test | 1.24 (0.98–1.52) | 1.46 (1.25–1.75) | 1.11 (0.93–1.45) | <0.001 |
| FEV1/FVC in bronchodilatation pre-test | 116.00 (116.00–118.00) | 116.00 (116.00–118.00) | 116.00 (116.00–118.00) | 0.4 |
| PEF in bronchodilatation pre-test | 3.89 (3.14–4.46) | 4.15 (3.58–4.48) | 3.57 (3.07–4.43) | 0.052 |
| FEF25-75 in bronchodilatation pre-test | 2.68 (2.39–3.40) | 2.78 (2.56–3.45) | 2.60 (2.27–3.35) | 0.2 |

²Chi-square test of independence; Fisher's exact test; Wilcoxon-Mann-Whitney test

N (%)or Median (IQR)

N if ≠ N*

### Children's perceptions and knowledge of pollution

**Drawn elements.** Eight children took part in the "Draw and express yourself" activity (Fig 2). Children's representation of air pollution was mainly reflected in the presence of pollution sources in their drawings such as household waste on the ground or in the water (drawing 1), presence of burning tires (drawing 2), waste on the ground (drawing 7 and 8), and presence of smoke (drawings 1, 3 and 5). The main consequences of pollution were represented as the reduction of life forms, with fewer trees and flowers on the polluted drawing (drawings 1, 5, 7 and 8). The presence of human activity was represented by changes in the type and size of building shown: small house on the unpolluted drawing, large house on the polluted drawing (drawing 1, 5 and 7).

**Additional elements through speech.** In their drawings, children associated air pollution with dirt, the presence of garbage and the presence of smoke. The notion of unpleasant odors was added to the discourse. The children draw comparisons between different types of infrastructure: the most imposing and city-like being the most polluted: *"City houses pollute much more than mud houses in villages"* (drawing 1), *"houses are replaced by buildings that pollute more"* (drawing 5), *"the house in the polluted drawing is a factory and the other a non-polluted shack"* (drawing 7). In addition, the notion of ventilation through house windows to keep the air unpolluted was expressed through speech (drawings 3 and 5). The notion of *"bad smells"* was also mentioned by the children during the oral explanation of their drawings (drawings 1, 5, 6 and 8). Half of children considered air pollution to be a serious problem: *"it can kill"* (speech 1), *"you can get sick"* (speech 2), *"people will get sick"* (speech 5). Other children were unable to answer the question (children 4 and 8), and others did not consider air pollution to be a serious health issue (speeches 3, 6 and 7). Finally, only two children explained that air pollution is serious for the environment: *"it can kill plants"* (speech 3), *"people will get sick, animals, vegetation"* (speech 5). Discussions revealed that burning fuel such as wood or coal, especially for cooking, was also a source of air pollution: *"If you burn wood, it pollutes"* (speech 1), *"Smoke from coal in the house"* (speech 3), *"Coal makes you smoke, it pollutes the air"* (speech 5).

The majority of children explained that they had heard about air pollution at school. One of the children mentioned having followed a course on pollution at school in second grade. This child produced the most changes on the drawing between the two landscapes, and was also able to provide the most sources, consequences and explanations on the subject orally.

## Discussion

Exposure to air pollution was high in poorly living district of Abidjan, with PM2.5 and PM10 concentrations exceeding WHO recommendations. Multiple and significant sources of air pollution were identified, related to living conditions and the environment. The prevalence of respiratory symptoms was high among children, and spirometry results showed a high prevalence of LFI and asthma. Children and adults only perceive visible sources of air pollution. A lack of information on the subject was noted in the interviews.

Indoor and outdoor $PM_{2.5}$ and $PM_{10}$ concentrations exceeded WHO-recommended thresholds of 45 μg/m$^3$ over 24h for $PM_{10}$, and 15 μg/m$^3$ over 24h for $PM_{2.5}$ [42]. Looking at other measurements taken in Abidjan, the median $PM_{10}$ and $PM_{2.5}$ concentrations are 45.9 μg/m$^3$ and 20.5 μg/m$^3$ respectively [48–50]. The concentrations of PM at the study site in Andokoi were similar to those recorded in 2016 at the same location [41]. The various sources of pollution identified in the neighborhood and by the population, whether children or their parents, help us to understand that these populations are exposed, by the geography of the neighborhood, by the type of housing and the characteristics of the dwellings, but also and above all

by cooking fires. These sources may explain why the Andokoi data are slightly higher than the city data and WHO recommendations. There is also little difference between pollution concentrations in homes and outdoors. This can be explained by the fact that homes are often open onto courtyards and have few rooms.

None of the households used biomass fuels exclusively. In comparison, during the 2016 Kouao study in the same Andokoi area, 15% of households used biomass fuel (wood or charcoal) exclusively for cooking [41]. Collecting annual data on PM concentrations in the neighborhood would enable us to identify whether the decrease in PM concentrations from 2016 to 2022 can be attributed to the change in the type of fuel used for cooking, or whether it depends on seasonality or other factors. Data from the adult interviews seem to confirm that coal is gradually being abandoned in favor of gas. It would be interesting to know the proportion of use between gas and coal for those using these two forms of fuel. Despite a change in cooking practices, pollutant levels remain high, which may explain the significant prevalence of respiratory disorders and symptoms among children.

Of the 124 children included, the prevalence of respiratory symptoms suggestive of asthma was high. The results are similar to those presented in the study of Kouao. In 2016, 19 children had wheezing in the last 12 months (18.3%) and 45 had night-time wheezing (43.3%). Similarly, 21.8% of parents reported a previous asthma attack, compared with 18.3% in 2016 [41]. The prevalence of chest wheezing is much higher in our study compared to other regions.

Table 4. Adults' perception of air pollution (N=59).

| Characteristics | N =*59 N (%) |
|---|---|
| Neighborhood polluted by air pollution | 40 (67.8) |
| Information on | |
| Air pollution | 33 (55.9) |
| The health effects of air pollution | 38 (64.4) |
| What to do about air pollution | 16 (27.1) |
| Health impacts of air pollution | |
| No consequences | 5 (8.5) |
| Few consequences | 8 (13.5) |
| Strong consequence | 36 (61.0) |
| Very high impact | 8 (13.6) |
| Don't know | 2 (3.4) |
| Impacts of air pollution on the environment | |
| No consequences | 7 (11.9) |
| Few consequences | 9 (15.2) |
| Strong consequence | 31 (52.5) |
| Very high impact | 6 (10.2) |
| Don't know | 6 (10.2) |
| Sources identified as fairly or highly polluting | |
| Industries/plants | 49 (83.0) |
| Dust | 47 (79.7) |
| Vehicles and transport | 45 (76.3) |
| Garbage burning | 41 (70.7) |
| Fish or meat smoking activity | 34 (57 .6) |
| Open combustion fires | 28 (47.5) |

*6 non-respondents

In Cameroon, in a study carried out in 2016, 182 pupils (7.6%) (CI 95% 6.5 - 8.6) had been wheezing for the previous 12 days [51]. In Djibouti, in a study carried out from 2014 to 2016, the prevalence of wheezing in the chest, wheezing on exertion and nocturnal dry cough in the ISAAC questionnaire were respectively 5.6% (CI 95% 4.3 - 6.4), 1.6% (CI 95% 0.8 - 1.9) and 7.8% (CI 95% 7.1 - 8.23) [52]. The presence of a LFI evaluated at respiratory function tests was very important. Due to the difficulty some children have in correctly performing spirometry

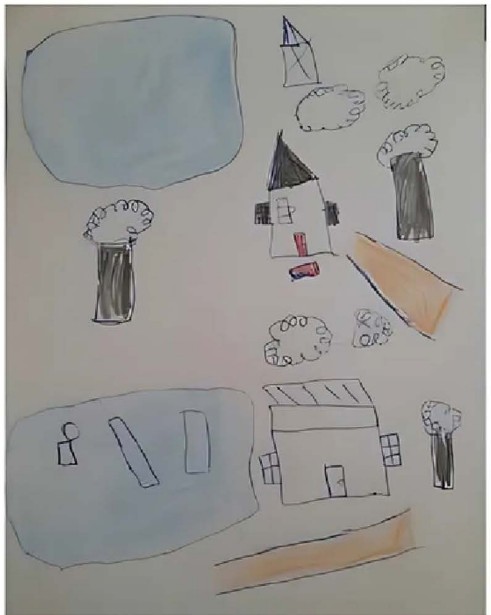
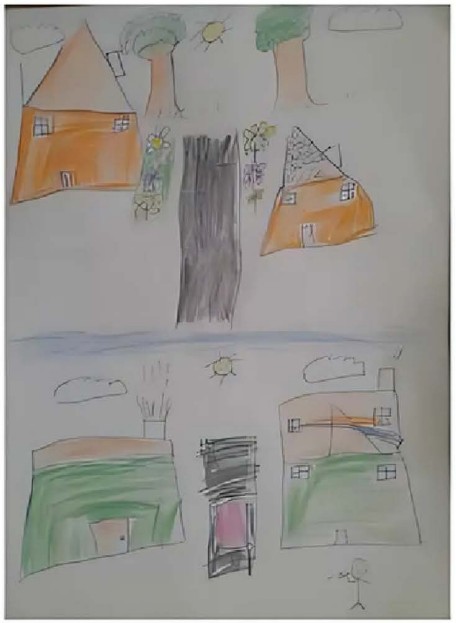
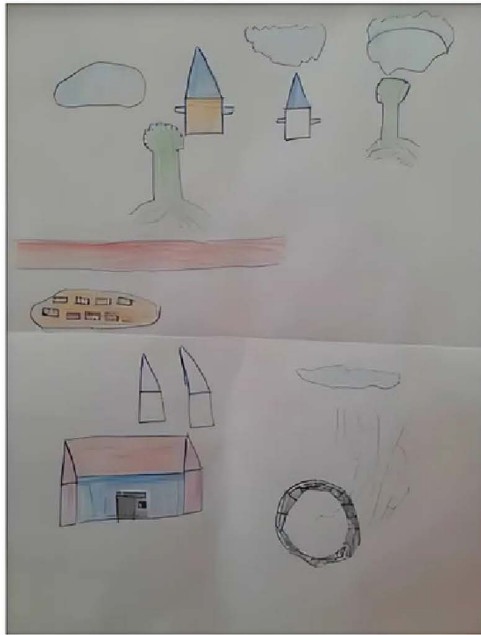
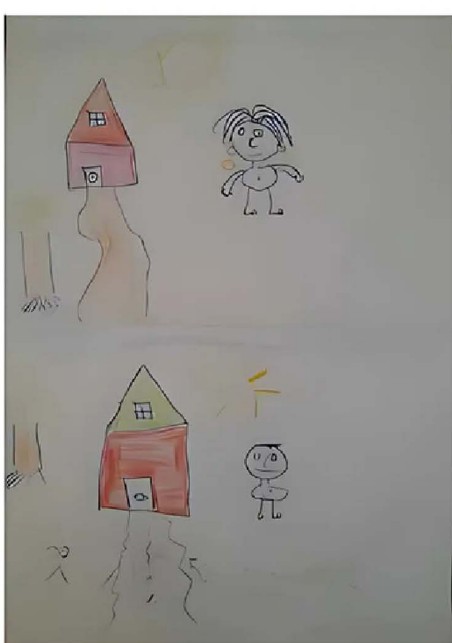

**Fig 2. Examples of children's work from the "Draw and express yourself" workshop, drawings 1, 2, 5 and 7, SEPol-CI, Andokoi, 2022.**

and because of the variability of spirometry curves at these ages, it would be preferable not to take into account spirometric data exclusively. Prevalence may be slightly overestimated. Spirometry requires a very specific technique, which may not be possible in a single session. Several sessions could be envisaged to further guarantee this bias. We found no association between exposure to various sources of pollution and the presence of LFI or wheezing in the last 12 months. One possible explanation is that our sample size is small.

The concept of air pollution remained fairly abstract for both children and adults interviewed in Yopougon. They associated air pollution with environmental pollution in general, and plastic pollution of water and soil in particular. Children cited, as in other studies, unpleasant odors and the presence of smoke as constitutive elements of air pollution, and the sources of pollution they most often cited were dust, biomass combustion and plastic waste, all visible elements [35,43,53]. In this study, below 8 years old, children were not able to express their perceptions on the subject in drawings or orally. Beyond the age, a study carried out in the USA in 2021 showed that parents' level of education influences teenagers' perception of air pollution [54]. Air pollution is not a source of concern for the population, which puts financial problems first. In a study carried out in Beirut in 2016, the deterioration of air quality was not cited as the main risk perceived by the population living in the suburb, who were mainly sensitive to risks related to civil security, road traffic and terrorism. Risks linked to pollution and environmental degradation only came in fourth place, almost equal with those linked to terrorism [55].

Both children and adults interviewed recognized an important role in human activities (transport, factories) and lifestyles (biomass combustion for cooking, garbage burning) as generating air pollution.

According to the interviews, public policies are not seen as a solution to the city's air pollution problems. Moreover, since biomass combustion fires are only considered a source of strong or very strong air pollutants by half of all women, this raises the question of the exposure of children who are carried on women's backs, while they're young, preparing meals. Few studies have focused on children's perceptions of air pollution in low- and middle-income countries, and it would be interesting to be able to compare these data with other neighborhoods or neighboring countries.

This study has several limitations. First, we restricted the study to households and children in one area of the commune of Yopougon, which is not representative of the whole commune, and of the whole city of Abidjan. A selection bias may also be present due to the fact that we did not have at our disposal the complete lists of households and children who participated in CHAIRPOL program in 2016-2017. Some households may therefore have been part of the previous study and be classified in our study as a new inclusion or vice versa. It is crucial to highlight the impact of seasonality on pollution and respiratory symptoms, particularly regarding the dry season, wet season, and the presence of Harmattan. In both CHAIRPOL and this study, these factors were not considered, and it will be essential to integrate them into future projects. The data collected may have been subject to recall bias, as we asked parents about their child's respiratory health from an early age. With regard to respiratory symptoms, the information gathered was not clinically verified. Prior knowledge of the exposure factors or pathologies concerned may influence the intensity of the search for exposure factors, or lead to further investigation of sick subjects. It would have been interesting to collect a certain amount of data on the respiratory health of parents to complement that of their children.

The strength of this work lies in its interdisciplinary and holistic approach, combining data from different perspectives and disciplines: public health, paediatric pulmonology, atmospheric physics and sociology. The different data sources were combined, using one type of data collected to compensate for the weakness of another, or to "check" the validity of the

results. The aim of this method is to increase the quality of the results, to improve understanding of the phenomenon studied and increase the reliability of the study, and finally to increase confidence in the validity of the results.

Using the validated ISAAC questionnaire, the children's respiratory health was assessed by determining the prevalence of bronchial symptoms suggestive of asthma, which were high in this population. Respiratory function test results confirmed the presence of ventilatory disorders and asthma in these children. Households are changing their habits in terms of the type of fuel used for cooking. Gas is gradually replacing the use of coal and wood, which is encouraging in terms of exposure to this source of pollution. Recommendations and awareness-raising for parents and children, starting at primary school level, are needed to limit the respiratory consequences for children of exposure to air pollution. In view of the results, schools would be the ideal medium for providing the most content on the subject of pollution, and air pollution in particular. The methodology implemented for this pilot project enabled us to gain a holistic view of the subject, and to draw lessons for future, larger-scale projects.

## Acknowledgments

The authors would like to thank the study participants, families and children for their participation, and students who carried out the data collection and analysis. They express their gratitude to Professor Michael Fayon and Doctor Ida Viho for their interest in this subject and their support and the PACCI research center for the logistics and technical implementation of the study.

## Author contributions

**Conceptualization:** Auriane Pajot, Madina Doumbia, Kouassi Kouao, Cathy Liousse, Raoul Moh, Joanna Orne-Gliemann, Flore Dick Amon Tanoh, Olivier Marcy, Véronique Yoboue.

**Data curation:** Auriane Pajot.

**Formal analysis:** Auriane Pajot.

**Funding acquisition:** Olivier Marcy.

**Investigation:** Auriane Pajot, Marie Yapo, Sarah Coulibaly, Stephane Ahoua, Sonia Adjoua Dje.

**Methodology:** Auriane Pajot, Joanna Orne-Gliemann, Olivier Marcy, Véronique Yoboue.

**Project administration:** Auriane Pajot, Raoul Moh, Olivier Marcy, Véronique Yoboue.

**Resources:** Auriane Pajot, Olivier Marcy.

**Software:** Auriane Pajot.

**Supervision:** Cathy Liousse, Olivier Marcy, Véronique Yoboue.

**Validation:** Madina Doumbia, Sylvain Gnamien, Cathy Liousse, Raoul Moh, Joanna Orne-Gliemann, Flore Dick Amon Tanoh, Olivier Marcy, Véronique Yoboue.

**Writing – original draft:** Auriane Pajot.

**Writing – review & editing:** Auriane Pajot.

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
