## [Decision Letter · Decision Letter 0]

14 Oct 2024

PGPH-D-24-01910

Air pollution exposure, respiratory consequences, and perceptions among urban African children living in poor conditions – a case study in Abidjan, Côte d'Ivoire

Dear Dr. Pajot,

Thank you for submitting your manuscript to PLOS Global Public Health. After careful consideration, we feel that it has merit but does not fully meet PLOS Global Public Health’s publication criteria as it currently stands. Therefore, we invite you to submit a revised version of the manuscript that addresses the points raised during the review process.

Please address comments regarding seasonal variation adjustments, improve data accuracy, increase the sample size, clarify the depth of interviews with children, and provide details regarding inconsistent sample sizes. Also, please address the grammatical issues and define abbreviations upon first use.

We look forward to receiving your revised manuscript.

Kind regards,

Giridhara R Babu, MBBS, MPH, PhD

Academic Editor

Journal Requirements:

**Please only choose the relevant sentences from below**

1. Please clarify all sources of funding (financial or material support) for your study. List the grants (with grant number) or organizations (with url) that supported your study, including funding received from your institution. 

2. State the initials, alongside each funding source, of each author to receive each grant.

3. State what role the funders took in the study. If the funders had no role in your study, please state: “The funders had no role in study design, data collection and analysis, decision to publish, or preparation of the manuscript.”

4. If any authors received a salary from any of your funders, please state which authors and which funders.

2. We note that your Data Availability Statement is currently as follows: "All processed data are available in the submitted article"

3. We ask that a manuscript source file is provided at Revision. Please upload your manuscript file as a .doc, .docx, .rtf or .tex.

4. Please provide separate figure files in .tif or .eps format.

Additional Editor Comments (if provided):

Reviewers' comments:

Reviewer's Responses to Questions

**Comments to the Author**

1. Does this manuscript meet PLOS Global Public Health’s publication criteria ? Is the manuscript technically sound, and do the data support the conclusions? The manuscript must describe methodologically and ethically rigorous research with conclusions that are appropriately drawn based on the data presented.

Reviewer #1: Yes

Reviewer #2: Yes

Reviewer #3: Yes

2. Has the statistical analysis been performed appropriately and rigorously?

Reviewer #1: Yes

Reviewer #2: Yes

Reviewer #3: Yes

3. Have the authors made all data underlying the findings in their manuscript fully available (please refer to the Data Availability Statement at the start of the manuscript PDF file)?

Reviewer #1: No

Reviewer #2: Yes

Reviewer #3: Yes

4. Is the manuscript presented in an intelligible fashion and written in standard English?

Reviewer #1: Yes

Reviewer #2: Yes

Reviewer #3: Yes

5. Review Comments to the Author

Reviewer #1: Reviewer’s report

Line 109: the study time was stipulated, but I think the different tests would have been conducted in different seasons in that area. This will enable accurate comparisons of the results. Equally, the study that was 2016 did not take into consideration the various seasons of the area

Line 115: I don’t think children can respond in-depth to an interview. We can say interviewing children aged 5-10

Line 154: you had a range of age done, say 5 and over, but remain within the range that was specified.

Line 182: correct the grammar in that sentence

Remark: When abbreviating words for the first time, define them

Line 214: correct the phrase home fish smoking fish

Line 221: how many interviews were conducted? Specify it

Line 259: remove one ‘’had’’

Line 339: you should be able to specify the number of people who raised the need for public awareness of pollution to be addressed

Line 356: remove one ‘’if they did’’

Remark: From lines 334 to 358, most of your observations came from interview 3. Why?

Personal views:

The work is quite rich and educative. I think the sample size can be increased and different seasons considered. It will improve the results of the next studies.

There was no similarity between youngsters residing in gas-powered homes and those whose primary fuel source was coal or firewood.

The Chai square was not adequately used to compare the variables (the Chi-Square test is used to determine whether there is a significant association between categorical variables). and you mentioned it in your methods.

Reviewer #2: The strength of this study lies in using mixed method approach and also of involving both parents and children to understand their perception of air pollution and it's effects.

However a small sample size and fewer examinations of spirometry and clinical assessments could be the reason for not finding any significant association with the risk factors.

Below are some of the comments to be addressed.

Provide Reference to this statement in introduction "In 2019, 99% of the world's population lived in areas where air pollutant concentration exceeds WHO recommended air quality thresholds."

Line 79: "high" burden

Line 104: "improve"

Line 172: respiratory symptom or asthma symptom not clear

Line 181: unable to "undergo"

Provide any references or justification for the cut-offs chosen for OVD, RVD and asthma.

Line 214: Tobacco

Line 214: what is "home fish smoking fish"?

Was the amount of time the child spends at home, presence of child during cooking activities, ventilation inside home assessed? These parameters could be important factors that need to be accounted for.

Table 1: Explain what is hevea? In table it is Hevea in the result section you mention rubberwood.

Type of cooking equipment: Explain the difference between traditional oven, improved cooker and gas cooker. Has stove type and fuel both been mixed in this variable?

Table 2 : replace size with height

Line 433: Incomplete sentence, state the numbers “The prevalence of wheezing in the chest during the year, a dry cough at night.”

Line 436: “These results are far superior to other studies”. Which results are you speaking about? The prevalence in your study for wheezing is much higher than in the Camerron study or Djibouti study. You can state “The prevalence is much higher in our study compared to other regions”

Line 441: “Because of the difficulty some children have in making spirometry curves correctly, and because of the variability of spirometry curves at these ages, it would be preferable not to take into account spirometric data exclusively.” Instead state “Due to the difficulty some children have in correctly performing spirometry and because of the variability of spirometry curves at these ages, it would be preferable not to take into account spirometric data exclusively.”

Please mention the correct sample size

Line 477: As our sample size is small (59 households and 117 children),

Line 231: We included 124 children from 65 households

Reviewer #3: I found the article engaging, particularly the qualitative section that explored perceptions of air pollution. The insights shared by the children regarding the impacts of air pollution were especially valuable. Here are my specific comments:

INTRODUCTION

Line 73: In 2016, approximately 543,000 deaths of children under 5 and 52,000 deaths of children aged 5 to 15 were attributed to household and outdoor air pollution.

Could be updated to - In 2021, more than 700,000 deaths in children under 5 years of age were from diseases linked to air pollution in 2021. Of this, over 500,000 deaths were linked to exposure to household air pollution. (IHME, Global burden of disease 2024)

Line 88: In Côte d'Ivoire, outdoor air pollution is mainly due to anthropogenic activities such as burning of dumped waste in open air, fish smoking activities..

Complete this sentence. What other anthropogenic activities are being referred to?

Line 103: This interdisciplinary work could help to understand pollution phenomena and find concrete solutions to protect exposed populations and improved their quality of life.

Rephrase this sentence. What is meant by pollution phenomena?

METHODS

DATA ANALYSIS

Quantitative data analysis

Line 214: ‘home fish smoking fish’

A bit confusing! Rephrase as ‘smoking fish/ meat in the house/ courtyard

Line 264: Table 2. Demographic and medical characteristics, clinical signs, vital parameters and prevalence of lung function impairment of children

Clinicals signs reported on the day of clinical examination

What does ‘Cough nation’ mean?

DISCUSSION

Line 495: Respiratory fonction test results confirmed the presence of ventilatory disorders and asthma in these children.

Correct ‘fonction’ to ‘function’

6. PLOS authors have the option to publish the peer review history of their article (what does this mean? ). If published, this will include your full peer review and any attached files.

**Do you want your identity to be public for this peer review?** For information about this choice, including consent withdrawal, please see our Privacy Policy .

Reviewer #1: No

Reviewer #2: **Yes: ** Deepa Ravi

Reviewer #3: No

---

## [Decision Letter · Decision Letter 1]

18 Mar 2025

Air pollution exposure, respiratory consequences, and perceptions among urban African children living in poor conditions – a case study in Abidjan, Côte d'Ivoire

PGPH-D-24-01910R1

Dear Mme Pajot,

We are pleased to inform you that your manuscript 'Air pollution exposure, respiratory consequences, and perceptions among urban African children living in poor conditions – a case study in Abidjan, Côte d'Ivoire' has been provisionally accepted for publication in PLOS Global Public Health.

Best regards,

Giridhara Rathnaiah Babu, MBBS, MPH, PhD

Academic Editor

Reviewer Comments (if any, and for reference):

Reviewer's Responses to Questions

**Comments to the Author**

1. If the authors have adequately addressed your comments raised in a previous round of review and you feel that this manuscript is now acceptable for publication, you may indicate that here to bypass the “Comments to the Author” section, enter your conflict of interest statement in the “Confidential to Editor” section, and submit your "Accept" recommendation.

Reviewer #1: All comments have been addressed

Reviewer #3: All comments have been addressed

2. Does this manuscript meet PLOS Global Public Health’s publication criteria ? Is the manuscript technically sound, and do the data support the conclusions? The manuscript must describe methodologically and ethically rigorous research with conclusions that are appropriately drawn based on the data presented.

Reviewer #1: Yes

Reviewer #3: Yes

3. Has the statistical analysis been performed appropriately and rigorously?

Reviewer #1: Yes

Reviewer #3: Yes

4. Have the authors made all data underlying the findings in their manuscript fully available (please refer to the Data Availability Statement at the start of the manuscript PDF file)?

Reviewer #1: Yes

Reviewer #3: Yes

5. Is the manuscript presented in an intelligible fashion and written in standard English?

Reviewer #1: Yes

Reviewer #3: Yes

6. Review Comments to the Author

Reviewer #1: They have actually addressed all the recommendations that were made. I, from my point of view, think it can be published

Reviewer #3: (No Response)

7. PLOS authors have the option to publish the peer review history of their article (what does this mean? ). If published, this will include your full peer review and any attached files.

**Do you want your identity to be public for this peer review?** For information about this choice, including consent withdrawal, please see our Privacy Policy .

Reviewer #1: No

Reviewer #3: No
